# $S$OFT$L$OC: ROBUST TEMPORAL LOCALIZATION UNDER LABEL MISALIGNMENT

## ABSTRACT

This work addresses the long-standing problem of robust event localization in the presence of temporally misaligned labels in the training data. We propose a novel versatile loss function that generalizes a number of training regimes from standard fully-supervised cross-entropy to count-based weakly-supervised learning. Unlike classical models which are constrained to strictly fit the annotations during training, our soft localization learning approach relaxes the reliance on the exact position of labels instead. Training with this new loss function exhibits strong robustness to temporal misalignment of labels, thus alleviating the burden of precise annotation of temporal sequences. We demonstrate state-of-the-art performance against standard benchmarks in a number of challenging experiments and further show that robustness to label noise is not achieved at the expense of raw performance.

## 1  INTRODUCTION

The surge of deep neural networks (LeCun et al., 2015; Schmidhuber, 2015) has accentuated the evergrowing need for large corpora of data (Banko & Brill, 2001; Halevy et al., 2009). The main bottleneck for the efficient creation of datasets remains the annotation process. Over the years, while new labeling paradigms have emerged to alleviate this issue (e.g., crowdsourcing (Deng et al., 2009) or external information sources (Abu-El-Haija et al., 2016)), these methods have also highlighted, and emphasized, the prevalence of *label noise*. Deep neural networks are unfortunately not immune to these perturbations as their intrinsic ability to memorize and learn label noise (Zhang et al., 2017) can be the cause of training robustness issues and poor generalization performance. In this context, the development of models robust to label noise is essential.

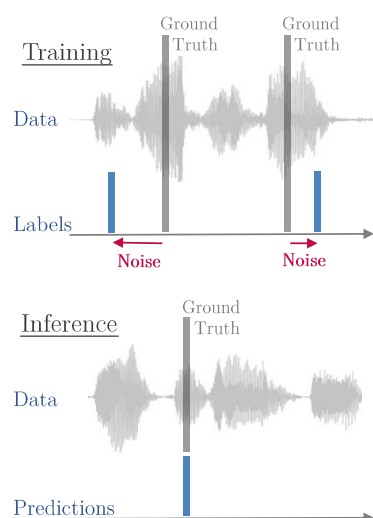

Figure 1: Temporal localization under label misalignment. Models are trained with noisy labels that differ from the actual ground-truth, while the final inference objective is the *precise* localization of events.

This work tackles the problem of precise temporal localization of events (i.e., determining when and which events occur) in sequential data (e.g. time series, video or audio sequences) despite only having access to poorly aligned annotations for training (see Figure 1). This task is characterized by the discrepancy between the precision required of the predictions during inference and the noisiness of the training labels. Indeed, while models are trained on inaccurate data, they are evaluated on their ability to predict event occurences as precisely as possible with respect to the ground-truth. In such a setting, effective models have to infer event locations more accurately than the labels they relied on for training. This requirement is particularly challenging for most classical approaches that are designed to learn localization by strictly mimicking the provided annotations. Indeed, as the training labels themselves do not accurately reflect the event location, focusing on replicating these unreliable patterns is incompatible with the overall objective of learning the actual ground-truth. These challenges highlight the need for more relaxed learning approaches that are less dependent on the exact location of labels for training.

The presence of temporal noise in localization tasks is ubiquitous given the continuous nature of the perturbation, in contrast to classification noise where only a fraction of the samples are misclassified. Temporal labeling is further characterized by an inevitable trade-off between annotation precision and time investment. For instance, while a coarse manual transcription of a minute of complex piano music might be achieved within a moderate time frame, a millisecond precision requirement — a common assumption for deep learning models — significantly increases the annotation burden. In this respect, models alleviating the need for costly annotations are key for a wide and efficient deployment of deep learning models in temporal localization applications.

This work introduces a novel model-agnostic loss function that relaxes the reliance of the learning process on the exact temporal location of the annotations. This softer learning approach inherently makes the model more robust to temporally misaligned labels.

**Contributions**   This work: a) proposes a novel loss function for robust temporal localization under label misalignment, b) presents a succinct analysis of the loss' properties, c) evaluates the robustness of state-of-the-art localization models to label misalignment, and d) demonstrates the effectiveness of the proposed approach in various experiments.

## 2   PROBLEM FORMULATION

The main assumption of this work is the instantaneous nature (i.e., lasting only one time-step in discrete time settings) of the events of interest. (Durations can nevertheless be modelled in such a framework by labeling the beginning and end of each event class as two separate channels.) Thus, for each sample, the ground-truth $\mathcal{T}^G := \{(t_m, c_m) \mid m \leq M\}$ consists of $M$ event occurrences each defined by its exact timestamp ($t_m \in \mathbb{R}_{\geq 0}$) and its class ($c_m \in [1, ..., d]$, with $d$ event classes). In this work, temporal label misalignment is then modelled by adding perturbations to the ground-truth timestamps:

$$\mathcal{T}^L := \{(t_m + \epsilon_m, c_m) \mid m \leq M\}, \text{ where } \epsilon_m \overset{iid}{\sim} E. \tag{1}$$

Although commonly defined as a normal distribution $\mathcal{N}(0, \sigma_i^2)$ (see experiments in Section 5.1 and 5.2), the noise distribution $E$ can also represent a wider range of perturbations (see experiments in Sections 5.2 and 5.3). The training dataset $\mathcal{D} := \{(\mathbf{X}_i, \mathcal{T}_i^L) : 0 < i \leq N\}$ is comprised of $N$ pairs with model input ($\mathbf{X}_i$) and misaligned labels ($\mathcal{T}_i^L$). The aim of this work is the following:

> **Objective**   *Estimate the true event occurrence times $\mathcal{T}^G$ of an unseen input sequence $\mathbf{X}$ using only the noisy data $\mathcal{D}$ for training.*

From a practical standpoint, although not necessary for the use of our loss, time is generally discretized. In such a discrete setting, each predictor $\mathbf{X}_i$ of the training data $\mathcal{D} := \{(\mathbf{X}_i, \mathbf{Y}_i) : 0 < i \leq N\}$ is an observable temporal sequence of length $T_i$ (i.e., $\mathbf{X}_i = (\mathbf{x}_i(t))_{t=1}^{T_i} \in \mathbb{R}^{T_i \times \lambda}$) such as a DNN-learned representation, a spectrogram or any other $\lambda$-dimensional time-series. The label sequence $\mathbf{Y}_i = (\mathbf{y}_i(t))_{t=1}^{T_i} \in \{0, 1\}^{T_i \times d}$ is then the discrete equivalent of $\mathcal{T}^L$. (Note that this last statement assumes that only one event per class can occur at each time-step; in cases where this assumption is violated, the use of smaller temporal granularity solves this issue.)

## 3   RELATED WORKS

**Temporal Localization Under Label Misalignment**   The literature on temporal noise robustness is limited despite the critical relevance of this issue. First, Yadati et al. (2018) propose solutions combining noisy and expert labels; however, unlike our approach, these methods require a sizable clean subset of annotations. Second, while Adams & Marlin (2017) achieve increased robustness by augmenting simple classifiers with an explicit probabilistic model of the noise structures, the effectiveness of the approach on more complex temporal models (e.g., LSTM) still needs to be demonstrated. Finally, Lea et al. (2017) perform robust temporal action segmentation by introducing an encoder-decoder architecture. However, the coarse temporal encoding comes at the expense of finer-grained temporal information, which is essential for the precise localization of short events (e.g., drum hits). In this paper, rather than a new architecture, we propose a novel and flexible loss function

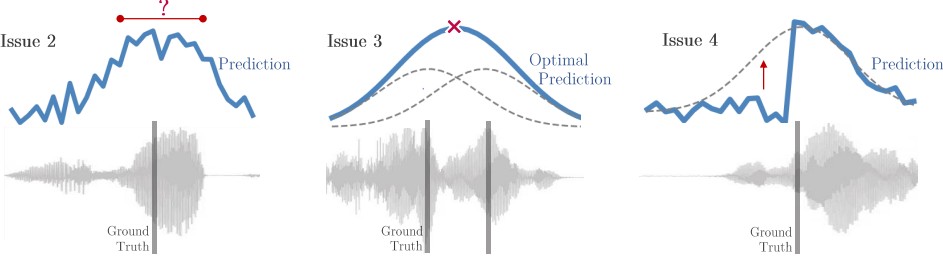

Figure 2: Inherent drawbacks of the classical trick of smoothing the labels *only*. **Issue 2**: ambiguous predictions of event locations require the use of additional heuristics. **Issue 3**: close events cannot be easily disentangled. **Issue 4**: the need to estimate the left-tail of label distributions compels the model to detect events before their actual occurrence, which might lead to overfitting.

— agnostic to the underlying network — which allows the robust training of temporal localization networks even in the presence of extensive label misalignment.

**Classical Heuristic**    Our approach is closely linked to the more classical trick of label smoothing or target smearing (e.g., applying a $\tilde{\sigma}^2$-Gaussian filter $\Phi_{\tilde{\sigma}^2}$ to the labels) which has been considered to increase robustness to temporal misalignment of annotations (Schlüter & Böck, 2014; Hawthorne et al., 2017). This slight modification of the input data converts the original point prediction problem into a distribution prediction problem. Indeed, the smoothing of the labels transforms the point labels into distributions. The algorithm is then trained to predict these distributions, which eventually have to be transformed back to point predictions using hand-crafted peak picking heuristics (see Figure 3 (*left*)). This methodology is also very common in 2D image keypoint detection applications which deal with spatial uncertainty, e.g. human pose estimation (Tompson et al., 2014; 2015) or facial landmark detection (Merget et al., 2018). However, despite its intuitive nature, this traditional solution presents several inherent drawbacks (see Figure 2): **(Issue 1)** Even in a noise-free setting, by transforming the impulse-like target into a distribution, the optimal model predictions (with respect to the training loss) differs from the actual goal of the pipeline (i.e., precise localization indicated by the original event label). **(Issue 2)** As the model learns by mimicking the smoothed target throughout the learning phase, the predictions themselves will be spread out over several time-steps. Hence, additional tailored heuristics, such as peak picking (Böck et al., 2013) or complex thresholding, are required to achieve precise temporal localization. **(Issue 3)** Even advanced peak picking approaches struggle to disentangle close events. For instance, a unique maximum might emergence in the middle of two events, thus significantly disturbing the timeliness of the final predictions. **(Issue 4)** Having the label mass dispersed temporally both before and after the event occurrences is problematic not only for causal models (i.e., models that make predictions at time $t$ only with data up to time $t-1$) but also for one-sided recurrent networks and fully convolutional architectures with limited receptive fields. Indeed, all these models have to estimate the left tail of the label distribution before even seeing the event occur. This requirement compels the model to find some structure before the actual event occurrence, leading to poor generalization performance. Although bidirectional networks do not suffer from it, this issue limits the range of possible architectures. The presence of strong label misalignment further worsens these four issues as increased noise commonly warrants increased smoothing, dispersing the label (and consequently prediction) mass even more. Overall, experimental evidence (e.g. Section 5.1.1) shows that the accumulation of these issues proves to be very detrimental to the noise robustness of these classical approaches.

In contrast, this work presents a novel paradigm for dealing with temporal label uncertainty. The main idea consists in directly inferring point predictions rather than resorting to distributions or heatmaps by fully integrating the modelling of the noise into the loss function. Such a direct approach allows for an end-to-end learning of localization without the need for additional hand-crafted components. In addition, the systematic and standalone loss function proposed in this regard (Section 4.2) not only solves all the above-mentioned issues but also scales well to extensive label misalignment.

**Weakly-Supervised Learning**    Some weakly-supervised models leverage weaker annotations to infer more fine-grained concepts. In such frameworks, noisy labels are implicitly bypassed by the use of higher-level labels — which are more invariant to perturbations. For instance, some works achieve object detection (Fergus et al., 2003; Bilen & Vedaldi, 2016) or temporal localization (Kumar

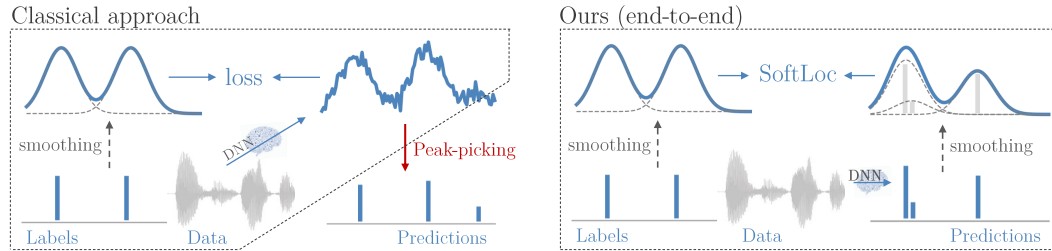

Figure 3: Modelling novelty. By smoothing *both* the labels and predictions, the model directly infers point predictions rather than distributions. Among other things, this modification allows for end-to-end learning of localization and alleviates the need for peak-picking. *(left)* Classical approach of smoothing the labels *only*. *(right)* Our novel end-to-end approach and its $\mathcal{S}$oftLoc loss function.

& Raj, 2016; Wang et al., 2017) using only class-level annotations. However, finer-grained labels, even noisy ones, often contain some additional information that is essential for optimal performance.

## 4 $\mathcal{S}$OFT$L$OC MODEL

### 4.1 SOFT LOCALIZATION LEARNING LOSS

The general principle of relaxing the localization learning is intuitive and potentially powerful if carefully implemented. However, by smoothing the label *only*, classical approaches transform the original point prediction problem into a distribution prediction problem which eventually causes issues (see Section 3). Many of the drawbacks arising from the asymmetric nature of the one-sided smoothing can however be alleviated by filtering not only the labels (i.e., $\Phi_{\mathcal{S}_i^2} * \mathbf{Y}_i(\cdot)$), but also the predictions (i.e., $\Phi_{\mathcal{S}_i^2} * \hat{\mathbf{Y}}_i(\cdot)$) with a unique softness parameter $\mathcal{s}_i$. The comparison of these two smoothed processes yields a relaxed loss function for the soft learning of the location that deals on its own with the temporal uncertainty of the labels. Indeed, in such a setting, the model is given input sequences of point-like events and directly infers point predictions without having to resort to distributions or heatmaps (see Figure 3 *(right)*); it is only the loss function that views these point labels and predictions as smoothed processes. In discrete time settings, the loss can be written as

$$\mathcal{L}_{\text{SLL}}(\theta) = \sum_i \mathcal{L}\big(\Phi_{\mathcal{S}_i^2} * \hat{\mathbf{y}}_{i,\theta}(\cdot), \Phi_{\mathcal{S}_i^2} * \mathbf{y}_i(\cdot)\big), \text{ with } \Phi_{\mathcal{S}_i^2} \text{ a } \mathcal{s}_i^2\text{-Gaussian filter,} \qquad (2)$$

where $\mathcal{L}(\cdot, \cdot)$ can be any measure of distance. The learning is characterized as *soft* since the loss is not strictly constraining in terms of precision or mass concentration. Indeed, the mass of each event can be both scattered over numerous time-steps and slightly shifted temporally without any abrupt increase in loss. Thus, the model's reliance on exact label locations is relaxed.

**Measure $\mathcal{L}$** In noise-free settings, the average stepwise cross-entropy is a common choice of loss function for state-of-the-art models (Lea et al., 2017; Wu et al., 2018; Hawthorne et al., 2019). While a potentially unbounded penalization of false predictions might be ideal when training on clean datasets, such behavior can be highly detrimental when labels are subject to temporal misalignment. Therefore, for all experiments in Section 5, $\mathcal{L}$ is set to the (bounded) average local mean-squared error.

**Properties** Symmetrically smoothing *both* the labels and predictions solves several of the issues highlighted in the previous section (see Figure 4). First, in a noise-free setting, the optimal predictions with respect to $\mathcal{L}_{\text{SLL}}$ are the original annotations themselves. **(Solves 1)**. Second, since the predictions are also smoothed over time, each trigger adds detection mass not only after, but also before the prediction time. Therefore, the model is not required the estimate the left-tail of the label distribution before the actual event occurrence **(Solves 4)**. The prediction mass for a particular event is not necessarily dispersed over time anymore. For instance, in noise-free settings, the point-like targets themselves are the solution to the optimization problem. However, $\mathcal{L}_{\text{SLL}}$ does not strictly constrain the mass of each event to be contained in a single time-step **(Partially Solves 2 & 3)**.

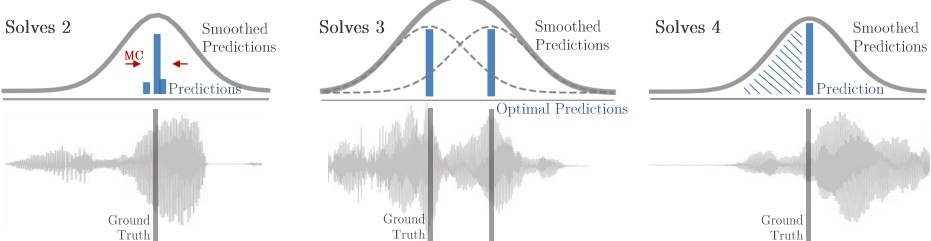

Figure 4: The different issues arising from only smoothing the labels are solved by our approach. **Solves 2**: the predictions are unambiguous as the model infers point predictions that are converged towards well-defined points in time. **Solves 3**: no disentanglement is required as the model directly infers point predictions instead of distributions. **Solves 4**: point predictions span symmetric distributions through smoothing, thus solving the left-tail estimation issue.

## 4.2 $\mathcal{S}$OFTLOC LOSS

The potential dispersion of the prediction mass and its direct consequences on localization performance still need to be addressed. To that end, we propose to leverage the properties of the weakly-supervised model defined in (Schroeter et al., 2019), which achieves precise temporal localization using only occurrence counts for training. Aside from exhibiting strong localization performance, the loss introduced in that work possesses an implicit mass convergence property, which concentrates the scattered prediction mass toward well-defined single points in time:

$$\mathcal{L}_{\mathrm{MC}}(\theta) = -\sum_i \log \left( \sum_{A \in F} \prod_{l \in A} \hat{y}_{i,\theta}(l) \prod_{j \in A^c} (1 - \hat{y}_{i,\theta}(j)) \right), \tag{3}$$

where $F$ is the set of all subsets of $\{1, 2, ..., T_i\}$ of size $\sum_k \mathbf{y}_i(k)$.

**Full $\mathcal{S}$oftLoc Model** Incorporating this mass convergence loss as a regularizer to our soft localization learning loss $\mathcal{L}_{\mathrm{SLL}}$ allows the model to directly achieve precise impulse-like localization, without weakening its noise robustness properties. Thus, this eliminates prediction ambiguity, as only a single point prediction is outputted per event occurrence **(Solves 2 & 3)**. Overall, when trained with the $\mathcal{S}$oftLoc loss,

$$\mathcal{L}_{\mathcal{S}\mathrm{oftLoc}}(\theta) = (1 - \alpha_\tau)\mathcal{L}_{\mathrm{SLL}}(\theta) + \alpha_\tau \mathcal{L}_{\mathrm{MC}}(\theta), \tag{4}$$

the model simultaneously softly learns to mimic the localization annotation, while converging the scatter mass toward impulse-like predictions. In this equation, $\alpha_\tau$ regulates the predominance of the mass convergence against the soft learning (for training iteration $\tau$). From a practical standpoint, starting with a moderate $\alpha_\tau$ allows an initial relaxed localization learning, before performing stronger mass convergence (see Section 5 for the specific settings used in this paper).

**End-to-end Learning of Localization** One of the key factors of the predominance of the deep learning models over classical ones relies on their ability to solve problems in an end-to-end fashion (Collobert et al., 2011; Krizhevsky et al., 2012), without the need to resort to partial optimization or hand-crafted heuristics. In contrast to more classical approach (see Section 3), our proposed method is an end-to-end solution to the problem of temporal localization in the presence of misaligned labels (see Issue 2). This solution eliminates the need for hand-crafted components (e.g. peak picking) and is expected to better serve the task at hand.

**Continuous Setting** While all experiments in Section 5 and most state-of-the-art temporal localization models perform a discretization of time, the loss definition can easily be adapted to suit continuous-time frameworks.

## 4.3 GENERALIZATION OF PAST WORKS

Our versatile $\mathcal{S}$oftLoc model is a generalization of several past works. Indeed, depending on the softness parameter $s_M$, the model encompasses a wide range of training regimes from classical fully-supervised to count-based weakly-supervised.

**Softness $\to 0$** By tending $s_M$ toward zero, the model becomes similar to a count-aware localization RNN with soft localization learning loss. For instance, setting $\mathcal{L}(\cdot) = -\log(1 - |\cdot|)$ yields

$$\lim_{S_M \to 0} \mathscr{L}_{\text{SLL}}(\theta) = -\sum_{i,t} \log\left(1 - |\hat{y}_{i,\theta}(t) - y_i(t)|\right)$$

$$\stackrel{y_i(t) \in \{0,1\}}{=} -\sum_{i,t} y_i(t) \log\left(\hat{y}_{i,\theta}(t)\right) + (1 - y_i(t)) \log\left(1 - \hat{y}_{i,\theta}(t)\right), \tag{5}$$

which corresponds to the sum of all stepwise cross-entropies. By further setting $\alpha_\tau = 0$ (i.e., discarding any count-awareness), our loss function becomes identical to the ones found in numerous temporal detection works (e.g., drum detection (Wu et al., 2018), piano onset detection (Hawthorne et al., 2017), and video action segmentation (Lea et al., 2017)).

**Softness $\to \infty$** Setting $s_M \to \infty$ causes the gradient of $\mathscr{L}_{\text{SLL}}(\theta)$ to vanish, discarding any prior information of localization, thus making the training weakly-supervised (Schroeter et al., 2019):

$$\lim_{S_M \to \infty} \mathscr{L}_{\mathcal{S}\text{oftLoc}}(\theta) = \alpha_\tau \cdot \mathscr{L}_{\text{MC}}(\theta) \propto \mathscr{L}_{\text{MC}}(\theta). \tag{6}$$

## 4.4 Dealing with Uncertainties

The introduced softness parameter can be leveraged to deal with different kinds of uncertainties. First, in contrast to the traditional approach of aggregating the annotations of multiple individuals (thus trading off dataset richness for noise reduction), our model can be trained on all conflicting individual sequences, since it can cope with noisy annotations. Second, an annotator specific softness $s_a^2$ can further be implemented to model their respective reliability. Finally, an extract specific softness can be incorporated to capture the noise or annotation complexity of certain more challenging sequences.

Experiments conducted in the section below show that the performance is robust to variations in the softness parameter. Indeed, this hyperparameter only acts as a coarse indicator of temporal uncertainty and thus does not need to strictly match the underlying noise distribution.

## 5 Experiments

In this section, we demonstrate the effectiveness and flexibility of our approach in a broad range of challenging experiments (music event detection, times series detection, video action segmentation). , *Experiment and implementation details can be found on the paper's website*[1].

### 5.1 Music Experiments

#### 5.1.1 Piano Onset Experiment

Piano transcription and more specifically piano onset detection is a difficult problem as it requires precise and simultaneous detection of hits from 88 different polyphonic channels.

**Dataset** This experiment is based on the MAPS database (Emiya et al., 2010). The dataset creation protocol strictly follows the one from Hawthorne et al. (2017). (Only onsets are considered for the comparison.) To evaluate the robustness, the training labels are artificially perturbed according to a normal distribution $\epsilon_m \sim \mathcal{N}(0, \sigma^2)$, while the test labels are kept intact for unbiased evaluation.

**Benchmarks** Three different benchmarks are considered. First, the state-of-the-art model (on clean data) proposed by Hawthorne et al. (2017) is highly representative of models aiming for optimal performance with little regard for annotation noise (*Hawthorne*). Second, a smoothed version of the first benchmark with extended onset length (i.e., over 96ms) illustrates the common practice used to achieve robustness (*Hawthorne (smoothed)*). Finally, as the first benchmark performs local classification using standard cross-entropy, the soft bootstrapping loss proposed by Reed et al. (2014) is leveraged instead for increased robustness (*Bootstrap (soft)*).

---

[1]Anonymous link: https://github.com/SoftLocICLR/submission

**Architecture, Training and Evaluation**    Our network is comprised of six convolutional layers (representation learning) followed by a 128-unit LSTM (temporal dependencies learning) and two fully-connected layers (prediction mapping). The network is trained using mel-spectrograms (Stevens et al., 1937) and their first derivatives stacked together as model input, while data augmentation in the form of sample rate variations is applied for increased robustness and performance. The loss (Equation 4) with softness $s_M = 100$ms is optimized using the Adam algorithm (Kingma & Ba, 2015). The models are evaluated on the *noise-free* test set using $F_1$-scores computed with the standard *mir_eval* library (Raffel et al.) and a 50ms tolerance (Hawthorne et al., 2017). ($\alpha_\tau = \max(\min(\frac{\tau - 10^5}{10^5}, .9), .2)$.)

**Results**    As depicted in Figure 5, our proposed $\mathcal{S}$oftLoc approach displays strong robustness against label misalignment; in contrast to all benchmarks, the performance appears almost invariant to the noise level. (See Appendix A.1 for discussion on the model's performance for $\sigma > 200$ms.) At $\sigma = 150$ms, only 26% of training labels lie within the 50ms tolerance. In this context, the score achieved by our $\mathcal{S}$oftLoc model (i.e., $\sim 75\%$) is unattainable for classical approaches, which do not take label uncertainty into account and attempt to strictly fit the noisy annotations. While standard tricks, such as label smoothing, slightly improve noise robustness (e.g., *Hawthorne (smoothed)*), their effectiveness is limited in contrast to our proposed approach. Finally, the parameters used throughout this experiment are fixed. However, as our loss is a strict generalization of the standard cross-entropy loss used by Hawthorne et al. (2017), the small performance gap for small noise levels can be reduced by setting $\alpha_\tau = 1$, $s_M^2 \to 0$ms and $\mathcal{L}(\cdot) = -\log(1 - |\cdot|)$.

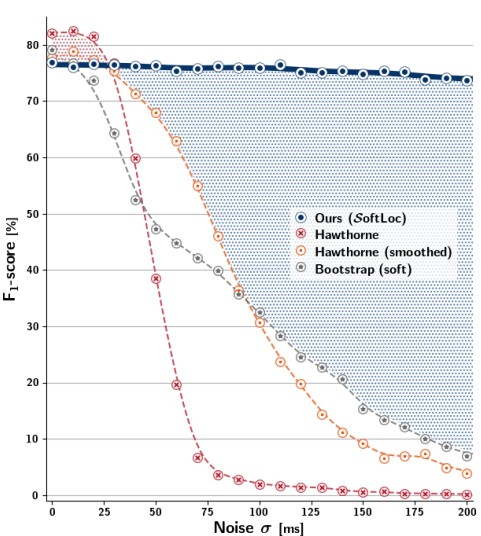

Figure 5: $F_1$ piano onset detection performance of our approach (softness $s_M = 100$ms) and the benchmark models as a function of label noise levels.

**Ablation Study**    To assess the usefulness of the different components of $\mathcal{L}_{\mathcal{S}\text{oftLoc}}$, we repeat the above experiments keeping only individual parts of the loss function. Table 1 reveals that $\mathcal{L}_{\text{SLL}}$ is the main driver of performance in noise-free settings, while $\mathcal{L}_{\text{MC}}$ ensures stability under increased label misalignment. (A simple threshold-based peak-picking algorithm was implemented to infer localization from the dispersed mass produced by $\mathcal{L}_{\text{SLL}}$.) Overall, while each loss individually produces reasonable predictions, only the combined $\mathcal{L}_{\mathcal{S}\text{oftLoc}}$ yields both competitive scores in noise-free settings and strong robustness to temporal misalignment.

### 5.1.2 DRUM DETECTION EXPERIMENT

The softness $s_M$ is a defining model hyperparameter. In this section, 210 independent runs for the same drum detection experiment are conducted with varying noise and softness levels in order to highlight the correlation between this key parameter, label noise and the final localization performance.

**Dataset**    The experiment is based on the D-DTD *Eval Random* drum detection task (IDMT-SMT-Drums dataset (Dittmar & Gärtner, 2014)) performed by Wu et al. (2018). The goal is the correct

Table 1: Ablation Study. Piano onset detection performance of our model trained with loss functions $\mathcal{L}_{\mathcal{S}\text{oftLoc}}$ ($s_M = 100$ms), $\mathcal{L}_{\text{SLL}}$ and $\mathcal{L}_{\text{MC}}$ respectively in various noise level settings.

| LOSS | $\sigma = 0$ms | 50ms | 100ms | 150ms | 200ms |
|---|---|---|---|---|---|
| $\mathcal{L}_{\text{SLL}}$ ($\alpha_\tau = 0$) | 76.06 | 76.00 | 75.10 | 66.88 | 46.91 |
| $\mathcal{L}_{\text{MC}}$ ($\alpha_\tau = 1$) | 71.59 | 73.04 | 68.69 | 70.33 | 67.26 |
| $\mathcal{L}_{\mathcal{S}\text{oftLoc}}$ | **76.88** | **76.34** | **75.86** | **74.87** | **73.68** |

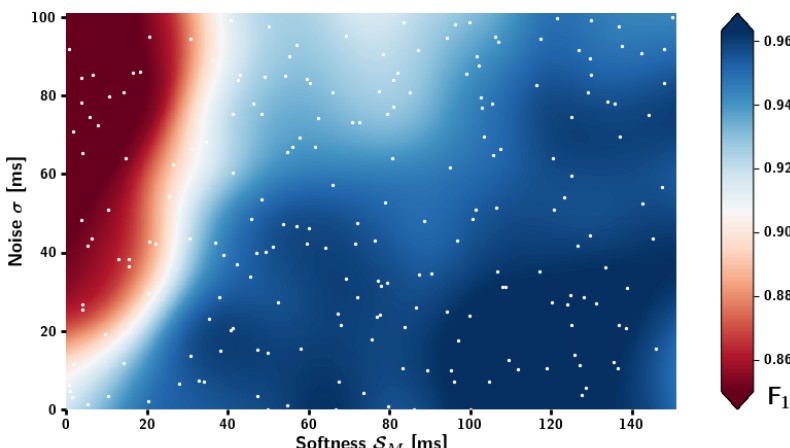

Figure 6: Drum detection performance with respect to model softness and label noise. $F_1$-scores are Gaussian Nadaraya–Watson estimates based on $210$ runs (white dots) sampled uniformly at random.

temporal localization of three different classes of drum hits — hi-hats (HH), kick drums (KD), and snare drums (SD) — within a $50$ms tolerance window. Normally distributed errors $\epsilon_m \sim \mathcal{N}(0, \sigma^2)$ are artificially introduced on all training and validation labels, while the test labels are kept intact for unbiased inference. The noise level $\sigma$ for each run is uniformly sampled from the range $[0\text{ms}, 100\text{ms}]$.

**Architecture, Training and Evaluation** The network is similar to the one in Section 5.1.1, except for the number of filters and nodes. The model softness $s_M$ for each run is uniformly sampled from $[0\text{ms}, 150\text{ms}]$. Training and evaluation are carried out in the same way as in the piano experiment in Section 5.1.1. *(Learning rate: $10^{-4}$, batch size: $32$, iterations: $1.5 \times 10^5$, sample length: $1.5$s)*.

**Results** The results of the $210$ runs are displayed in Figure 6. A Gaussian Nadaraya–Watson kernel regression (Nadaraya, 1964; Watson, 1964) is used to interpolate the $F_1$-score, offering a detailed view of the model's response to varying label noise levels. This figure not only confirms the model's high robustness to label misalignments, but also reveals that these results are very robust to changes in the softness level. Indeed, a wide range of softnesses yield optimal performance, as long as $s_M \geq \sigma$. Obviously, extreme softness levels (e.g. $s_M^2 \to \infty$) would however induce a partial or even total loss of the information conveyed by the localization prior, resulting in a decrease in performance (see Table 2). Robustness considerations aside, our $\mathcal{S}$oftLoc model displays an outstanding overall performance with $F_1$-scores over $95\%$ across all noise levels; the model — even when trained on extremely noisy labels (e.g., $\sigma = 100\text{ms}$) — outperforms several standard benchmarks Wu et al. (2018) which were trained on noise-free training samples ($\sigma = 0\text{ms}$).

**Noise-free Comparison** In clean settings (i.e., $\sigma = 0\text{ms}$), the benchmark models have a clear advantage as they correctly assume noise-free labels. Despite this, our $\mathcal{S}$oftLoc model achieves state-of-the-art performance on three different metrics (KD, HH, precision) demonstrating that robustness does not come at the expense of raw localization performance (see Table 2).

## 5.2 TIME SERIES DETECTION

The timely detection of events in healthcare time series is a crucial challenge to improve medical decision making. The task tackled in this section consists in the precise temporal detection of smoking episodes using wearable sensors features based on the puffMarker dataset (Saleheen et al., 2015). Once again, in order to conduct the robustness analysis, the original annotations are artificially misaligned. However, as each time-step in this dataset represents a full respiration cycle, the noise distributions must be applied in a discrete fashion: namely, rounded normal distribution (i.e., $E_i \sim \lfloor \mathcal{N}(0, \sigma^2) \rceil$) or binary constant length shifting of labels ($\delta$ steps either to the left or the right with equal probability), denoted $\mathcal{B}(-\delta, \delta)$. This task is particularly challenging as detections have to be perfectly aligned with the ground-truth to be considered correct.

**Model and Benchmark** As the focus is set on robustness rather than raw performance, the model architecture is kept extremely simple: a 14-node fully connected layer followed by a 14-unit LSTM

Table 2: *Noise-free* Drum Detection. Comparison of our $\mathcal{S}$oftLoc model ($s_M = 100$ms) and state-of-the-art models evaluated in Wu et al. (2018) on the clean D-DTD *Eval Random* task ($\sigma = 0$ms). The $F_1$-scores per instrument (KD/SD/HH), the average precision, recall, and overall $F_1$ are displayed.

| METHOD | KD | SD | HH | PRE | REC | $F_1$ |
|---|---|---|---|---|---|---|
| RNN | 97.2 | 92.9 | 97.3 | 95.7 | 96.9 | 95.8 |
| TANHB | 95.4 | 93.1 | 97.3 | 93.9 | 97.1 | 95.3 |
| RELUTS | 86.6 | 93.9 | 97.7 | 92.7 | 95.0 | 92.7 |
| LSTMPB | 98.4 | **96.7** | 97.4 | 97.7 | **97.6** | **97.5** |
| GRUTS | 91.4 | 93.2 | 96.2 | 91.8 | 97.2 | 93.6 |
| $s_M \to \infty$ | 96.0 | 90.4 | 97.1 | 95.1 | 93.9 | 94.5 |
| $s_M = 100$ms | *98.6* | 95.7 | *97.8* | *98.3* | 97.2 | *97.4* |

and a final fully connected layer with softmax activation. Both the standard cross-entropy (CE) and our $\mathscr{L}_{\mathcal{S}\text{oftLoc}}$ loss function are evaluated. The LR-M model proposed by Adams & Marlin (2017), which was developed to achieve strong robustness to temporal misalignment of labels on this particular dataset, is also considered as benchmark.

**Results** The results, produced using ten 6-fold (leave-one-patient-out) cross-validation. are summarized in Table 3. Not only does training with the proposed $\mathscr{L}_{\mathcal{S}\text{oftLoc}}$ loss function yield a strong improvement in robustness when compared to the standard cross-entropy, but our simple recurrent model also significantly outperforms the robust LR-M model on all metrics. In addition, our approach displays low standard deviations, which underlines the consistency and robustness of the learning. These observations hold for both noise distributions ($\mathcal{N}$ and $\mathcal{B}$); hence, the normal smoothing filters do not require the underlying noise to be normally distributed in order for the model to be effective. Further testing with skew normal distribution of noise confirm these results even in non-symmetric settings.

## 5.3 VIDEO ACTION SEGMENTATION

Video action segmentation — a dense classification problem where each time-step has to be mapped to one action class — differs substantially from music event localization or time series detection problems, where scattered events from multiple classes have to be precisely localized. Nonetheless, the properties of the $\mathcal{S}$oftLoc loss can still be leveraged on such a task; in this context, while the role of $\mathscr{L}_{\text{SLL}}$ is unchanged, $\mathscr{L}_{\text{MC}}$ acts as a count-based regularizer, rather than a means for mass convergence.

**Experiments** Several video segmentation experiments from Lea et al. (2017) are replicated using either the standard cross-entropy (original loss), $\mathscr{L}_{\text{SLL}}$ or $\mathscr{L}_{\mathcal{S}\text{oftLoc}}$ as training loss for the ED-TCN model. As the ED-TCN model already exhibits strong robustness properties against label misalignment Lea et al. (2017), these experiments will allow to measure the additional marginal gain in performance and robustness when replacing the standard cross-entropy with our the proposed $\mathscr{L}_{\mathcal{S}\text{oftLoc}}$ loss function. To assess robustness, each label sequence in the training set is either delayed or advanced by a fixed constant $\delta$. ($s_M = 7$s).

Table 3: Smoking Puff Detection. Comparison of LR-M (Adams & Marlin, 2017) and the deep model trained with CE or $\mathscr{L}_{\mathcal{S}\text{oftLoc}}$ with respect to misalignment distributions $\lfloor \mathcal{N}(0, \sigma^2) \rceil$ and $\mathcal{B}(-\delta, \delta)$. Reported metrics are mean and standard deviation of ten 6-fold cross-validated $F_1$-scores.

| | | $\sigma, \delta = 0$ | 1 | 2 | 3 | 4 |
|---|---|---|---|---|---|---|
| | LR-M | 93.0 (3.2) | 80.6 (8.6) | 65.9 (17.4) | 64.0 (15.6) | 55.0 (19.7) |
| $\mathcal{N}$ | CE | 92.6 (2.9) | 55.3 (16.2) | 36.0 (15.6) | 28.9 (17.0) | 25.8 (16.2) |
| | $\mathscr{L}_{\mathcal{S}\text{oftLoc}}$ | *93.1* (2.5) | *90.6* (3.4) | *87.8* (4.1) | *83.6* (5.2) | *79.0* (6.9) |
| | LR-M | — | 65.5 (14.5) | 54.9 (20.4) | 44.1 (19.7) | 51.8 (19.8) |
| $\mathcal{B}$ | CE | — | 41.7 (15.3) | 28.3 (14.5) | 26.6 (15.3) | 22.8 (15.1) |
| | $\mathscr{L}_{\mathcal{S}\text{oftLoc}}$ | — | *90.8* (3.3) | *87.0* (4.7) | *81.7* (7.2) | *72.4* (10.1) |

Table 4: Video Action Segmentation. Comparison of various training losses (CE, $\mathscr{L}_{\text{SLL}}$ and $\mathscr{L}_{\mathcal{S}\text{oftLoc}}$) with respect to different label misalignment levels for the ED-TCN model on 50 Salads (mid). Metrics are mean and standard deviation $F_1$@10 (Lea et al., 2017) of ten 5-fold cross-validations.

| Loss | $\delta = 0$s | 5s | 10s | 15s | 20s |
|---|---|---|---|---|---|
| CE | 66.7 (1.6) | 59.7 (1.2) | 43.4 (0.9) | 33.6 (1.1) | 26.7 (0.8) |
| $\mathscr{L}_{\text{SLL}}$ | 66.7 (1.0) | 60.6 (0.9) | 47.5 (1.2) | 36.1 (0.8) | 28.0 (1.2) |
| $\mathscr{L}_{\mathcal{S}\text{oftLoc}}$ | **67.2** (0.8) | **61.5** (1.3) | **48.0** (1.0) | **38.0** (1.9) | **29.5** (1.1) |

**Results**   As summarized in Table 4 and Table 5 (in Appendix B.2), replacing the standard cross-entropy loss with $\mathscr{L}_{\mathcal{S}\text{oftLoc}}$ does not only significantly increase the robustness of the ED-TCN model — which was already shown to be robust to label misalignment (Lea et al., 2017) — but also achieves competitive performance in noise-free settings. Further experiments with different softness parameters (see Figure 10 in Appendix B.1) reveal that increasing the model softness $s_M$ as the underlying noise levels increase produces optimal performance. For instance, in noisy settings, greater performance can be achieved (up to 25% overperformance) by simply choosing a large enough softness. Overall, the $\mathcal{S}$oftLoc loss function displays strong results on a very different task (i.e., temporal segmentation as opposed to temporal localization), highlighting once again its versatility of application.

## 6   Conclusion

In this work, we have shown how relaxing annotation requirements (i.e., weakening the model's reliance on the exact location of events) not only has the practical benefit of alleviating annotation efforts but, more importantly, leads to a model that is robust to temporal noise without compromising performance on clean training data. This contrasts with traditional approaches which attempt to strictly mimic the annotations, leading to poor predictions when training with noisy labels. We have demonstrated these claims on a number of classical challenging tasks, in which our $\mathcal{S}$oftLoc loss exhibits state-of-the-art performance.

The proposed loss function is agnostic to the underlying network and hence can be used as a loss replacement in almost any recurrent architecture. The versatility of the model can find applications in a wide array of tasks, even beyond temporal localization.

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

## A  PIANO ONSET DETECTION

### A.1  EXTREME NOISE SETTINGS

Figure 1 (in the main text) depicts the strong invariance of our $\mathcal{S}$oftLoc model to label misalignment on a broad array of noise levels (i.e., up to $\sigma = 200$ms). In this section, we evaluate the model's performance on an even wider range in order to fully assess its behavior in extreme settings. To that end additional piano onset detection experiments, with noise levels up to $\sigma = 1000$ms, were conducted following the protocol described in Section 5.1. The results are displayed in Figure 7.

Overall, this figure confirms the remarkable robustness of our $\mathcal{S}$oftLoc model to label misalignment. While the absolute performance unsurprisingly decreases as the training data becomes less accurate, the detection capability of the model in noisy settings outshines any classical approach (see Figure 1 in the main text). Finally, these results could further be improved by increasing the model softness $s_M$ (see Section 5.2).

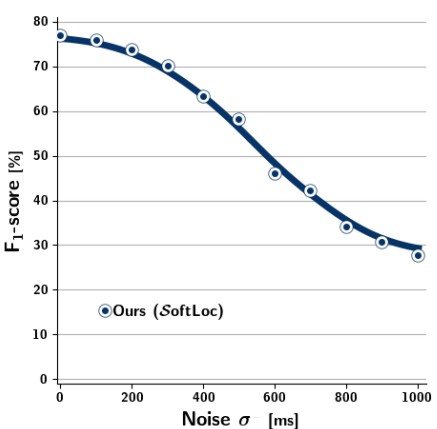

Figure 7: $F_1$ piano onset detection performance of the $\mathcal{S}$oftLoc model ($s_M\!=\!100$ms) as a function of label misalignment.

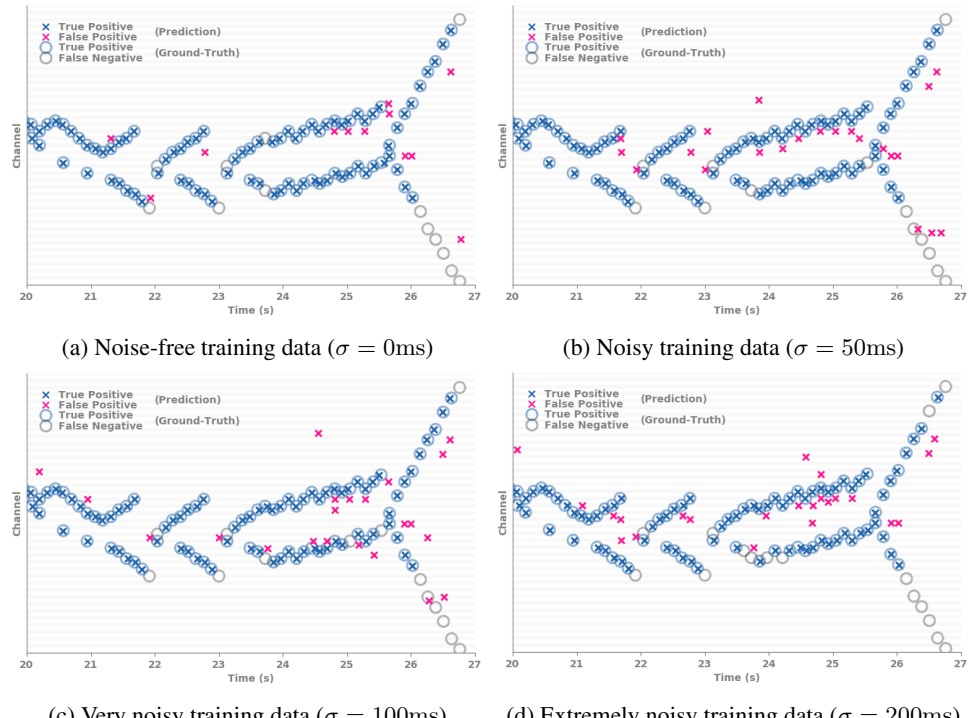

(a) Noise-free training data ($\sigma = 0$ms)  (b) Noisy training data ($\sigma = 50$ms)

(c) Very noisy training data ($\sigma = 100$ms)  (d) Extremely noisy training data ($\sigma = 200$ms)

Figure 8: Out-of-sample predictions of our $\mathcal{S}$oftLoc model trained on data subject to various levels of noise, ranging from (a) the noise-free case $\sigma = 0$ms to (d) the extremely noisy $\sigma = 200$ms. *(Schubert – Piano Sonata in A minor, D 784, Opus 143, 3. Mov)*

## A.2 FURTHER ILLUSTRATIONS

**Timeliness of $\mathcal{S}$oftLoc predictions** Figure 8 illustrates how consistently precise and well-centered (i.e., neither too late nor early) the predictions are regardless of the noise setting. Indeed, there is almost no difference in prediction centering when comparing the results for $\sigma = 0$ms or $\sigma = 200$ms.

**Noisy Labels and Ground-Truth Discrepancy** To further illustrate the complexity of the localization task when annotations are subject to misalignment, we consider the training labels as predictions and then compare them to the clean ground-truth. Figure 9 displays an example of the quality of the training labels. Obviously, in the noise-free setting (i.e., $\sigma = 0$ms), the localization is spotless as the training labels and the ground-truths are identical. However, as the noise level increases, the proportion of labels that stay within the 50ms tolerance window decreases significantly. More precisely, the performance (i.e., $F_1$-score) of the labels themselves is 68.2%, 39.8% and 23.7% for $\sigma$ equal to 50ms, 100ms and 200ms respectively.

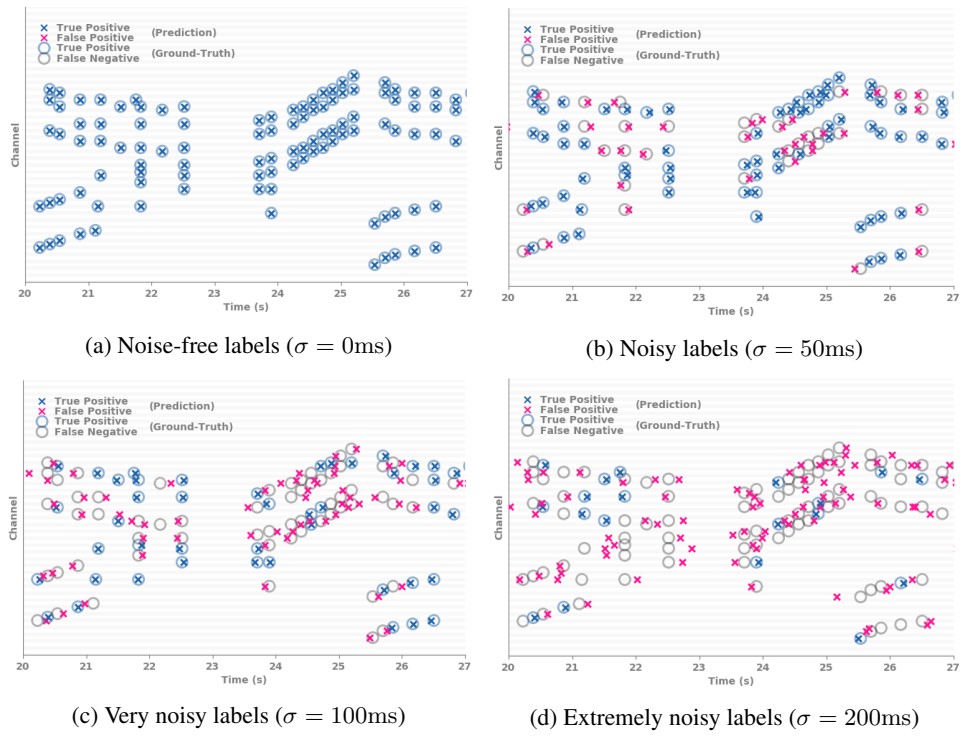

(a) Noise-free labels ($\sigma = 0$ms)          (b) Noisy labels ($\sigma = 50$ms)

(c) Very noisy labels ($\sigma = 100$ms)        (d) Extremely noisy labels ($\sigma = 200$ms)

Figure 9: In-sample performance of the noisy training labels themselves (*as predictions*) when compared to the clean ground-truth. *(Liszt – Hungarian Rhapsody No. 10)*

## B    VIDEO ACTION SEGMENTATION

### B.1    IMPACT OF THE SOFTNESS PARAMETER

As depicted in Figure 10, training with the $\mathcal{S}$oftLoc loss function instead of the standard cross-entropy yields improved performance (up to $25\%$) in all noise settings almost regardless of the softness $s_M$. The only exception occurs when selecting a softness level that is too wide while training with noise-free ($\delta = 0$) labels. As also observed in Section 5.1.2, the model achieves optimal performance when the softness level $s_M$ is slightly larger than noise level $\delta$. However, although the efficiency of the approach is bound to decrease when the disparity between selected softness and noise level is becoming too large, a performance close to the optimal one can be achieve with a wide range of softnesses $s_M$.

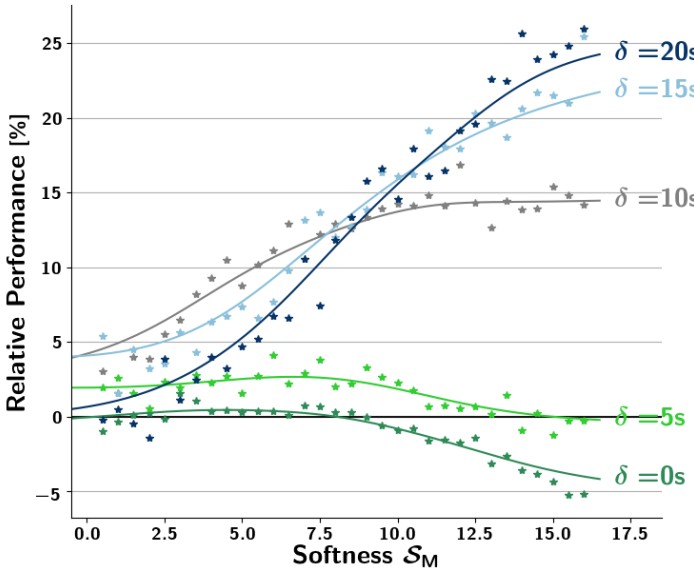

Figure 10: Video Action Segmentation. Relative performance of the ED-TCN model trained with $\mathcal{L}_{\mathcal{S}\mathrm{oftLoc}}$ — relative to CE — with respect to the softness level $s_M$ for various noise levels $\delta$.

## B.2 ADDITIONAL RESULTS

Table 5: Video Action Segmentation. Performance comparison of different training losses (cross-entropy, $\mathcal{L}_{\text{SLL}}$ and $\mathcal{L}_{\mathcal{S}\text{oftLoc}}$) for the ED-TCN model on various datasets and measures. Metrics are mean and standard deviation F$_1$@10 or F$_1$@50 (Lea et al., 2017) of ten 5-fold cross-validation.

|  |  | 50 SALADS (MID) | | | | |
|---|---|---|---|---|---|---|
|  | LOSS | $\delta = 0$s | 5s | 10s | 15s | 20s |
| F$_1$@10 | CE | **51.8** (0.7) | 38.5 (1.1) | 19.7 (0.7) | 10.7 (0.8) | 6.9 (1.0) |
|  | $\mathcal{L}_{\text{SLL}}$ | 50.7 (0.5) | 38.8 (1.4) | 22.3 (1.1) | 12.3 (0.8) | **7.8** (0.9) |
|  | $\mathcal{L}_{\mathcal{S}\text{oftLoc}}$ | *49.8* (0.9) | ***39.5*** (1.2) | ***23.4*** (1.4) | ***13.7*** (0.9) | *7.5* (0.8) |

|  |  | 50 SALADS (EVAL) | | | | |
|---|---|---|---|---|---|---|
|  | LOSS | $\delta = 0$s | 5s | 10s | 15s | 20s |
| F$_1$@10 | CE | 74.9 (0.8) | 72.7 (0.9) | 59.6 (1.2) | 48.1 (1.0) | 41.8 (0.6) |
|  | $\mathcal{L}_{\text{SLL}}$ | 75.4 (0.6) | 73.0 (0.9) | 61.9 (0.8) | 49.2 (0.9) | 41.8 (1.2) |
|  | $\mathcal{L}_{\mathcal{S}\text{oftLoc}}$ | ***75.5*** (1.2) | ***73.7*** (1.3) | ***62.7*** (1.5) | ***50.4*** (0.8) | ***43.8*** (1.7) |
| F$_1$@50 | CE | 63.2 (0.7) | 52.7 (1.5) | 34.4 (0.8) | 21.9 (1.4) | 15.3 (1.0) |
|  | $\mathcal{L}_{\text{SLL}}$ | 63.4 (1.0) | 55.3 (1.1) | 35.1 (1.0) | 22.8 (1.2) | 15.7 (1.2) |
|  | $\mathcal{L}_{\mathcal{S}\text{oftLoc}}$ | ***63.5*** (1.1) | ***55.7*** (0.9) | ***36.8*** (1.1) | ***23.6*** (0.9) | ***16.2*** (1.3) |

|  |  | GTEA DATASET | | | | |
|---|---|---|---|---|---|---|
|  | LOSS | $\delta = 0$s | 5s | 10s | 15s | 20s |
| F$_1$@10 | CE | **74.7** (1.2) | 64.3 (0.9) | 37.1 (1.9) | 27.6 (1.4) | **23.8** (1.6) |
|  | $\mathcal{L}_{\text{SLL}}$ | 74.1 (1.3) | 64.3 (2.4) | 41.3 (2.0) | 28.5 (1.4) | 22.8 (1.7) |
|  | $\mathcal{L}_{\mathcal{S}\text{oftLoc}}$ | *73.4* (1.2) | ***65.2*** (1.2) | ***43.8*** (2.7) | ***28.5*** (1.6) | *22.5* (0.8) |
| F$_1$@50 | CE | **59.3** (1.8) | 33.0 (1.9) | 12.0 (1.0) | 8.1 (0.9) | **7.8** (1.2) |
|  | $\mathcal{L}_{\text{SLL}}$ | 54.5 (1.5) | 33.7 (2.7) | 14.3 (1.0) | 8.0 (0.8) | 5.9 (1.0) |
|  | $\mathcal{L}_{\mathcal{S}\text{oftLoc}}$ | *52.0* (1.2) | ***34.8*** (1.2) | ***15.4*** (1.3) | ***8.4*** (1.4) | *5.1* (0.5) |

