# OpenReview forum: "SoftLoc: Robust Temporal Localization under Label Misalignment"
_ICLR.cc/2020/Conference — Reject_

### Official Review · AnonReviewer3 · 2019-10-23
**Official Blind Review #3**

**Rating:** 3

**Review:**

This paper proposes a new loss for training models that predict where events occur in a sequence when the training sequence has noisy labels. The central idea is to smooth the label sequence and prediction sequence and compare these rather than to force the model to treat all errors as equally serious.

The proposed problem seems sensible, and the method is a reasonable approach. The evaluations are carried out on a variety of different tasks (piano onset detection, drum detection, smoking detection, video action segmentation).

Suggestions
* While the authors do bring up lots of related work on learning from noisy labels, the insights from that work, and its relationship to this proposed technique could be more productively explored
* The connections between the assumptions of the evaluation metric and the motivation for the smoothing methodology could be more productively elucidated
* The task explored in this paper, and the task-specific problem, should be described more generally since ICLR has a generalist readership. For instance, the paper gets off on a rather strange footing discussing large data (indeed, but evident to the vast majority of ICLR readers), but little is said in terms of the specifics of the temporal localisation problem except via citations to other papers. In particular, the task set up, the problem posed by label misalignment could be described elegantly right in the introduction with a carefully designed figure. In section 2, the description is hard to follow, the mathematical notation is vague and hard to parse. For instance, the label sequence Y, is d-dimensional- the meaning of d should be clarified. Calling it “discretised” is also a bit strange, for most readers- it’s a label sequence.

The evaluation objective needs to be clarified, at least qualitatively; ideally in the introduction. Introducing a new objective is meaningful not only in light of noisy labels (always a problem), but in light of how the evaluation is carried out.



**Experience Assessment:**

I have published one or two papers in this area.

**Review Assessment: Checking Correctness Of Derivations And Theory:**

I assessed the sensibility of the derivations and theory.

**Review Assessment: Checking Correctness Of Experiments:**

I assessed the sensibility of the experiments.

**Review Assessment: Thoroughness In Paper Reading:**

I read the paper at least twice and used my best judgement in assessing the paper.

---

> ### Author Response · Authors · 2019-11-10
> **Response to Reviewer 3**
>
>
>
> Thank you for your insightful remarks. Based on your suggestions, Sections 1, 3 and 4.1 have been thoroughly updated and improved. For instance, the Introduction focuses now more on task specific matters, the Related works is more proactive, and the methodology is more clearly motivated. Figures 1 and 3 have also been added respectively to better illustrate the task at hand and to display the main conceptual differences with the classical approaches.
>
>
> (1) Related Works
> We agree with your assessment and have rewritten parts of the Related Works section to more proactively explain the differences between past works and our proposed approach. Figure 3 has also been added to more clearly illustrate the conceptual novelties. In addition, we took out the paragraph about classification noise, as the two problems are not closely related and setting the focus on a difference task might dilute the message about the objective of this work. Overall, we feel that these changes make the section more direct and proactive.
>
>
> (2) Methodology
> The chosen smoothing methodology builds on top of the intuitive classical approach of smoothing the labels only. The design choices are motivated by the need to improve these classical approaches. In the updated version of the text, Sections 3 and 4.1 as well as the new Figures 2, 3 and 4 highlight more clearly the issues caused by these widespread classical methods as well as the solutions provided by the proposed loss function. Overall, as the general motivation underlying the new methodology is mainly based on intuitive concepts, the paper presents a large experimental evaluation with 4 different experiments in 3 different domains (audio sequences, wearable sensors timeseries, video) to support the main claim of the work. The results confirm the effectiveness of the newly introduced loss function.
>
>
> (3) Introduction
> We agree that the specific problem addressed in this work was introduced too briefly in the initial version of the paper. Therefore, Figure 1 has been added to the first page to have a direct visual illustration of the task addressed in this work. Additionally, a full paragraph (2) focusing on the temporal localization task has been included in the updated text.
> While being trivial for the majority of the ICLR readership, we feel that the initial paragraph might be useful for reader coming from a more general audience.  Indeed, the first few sentences gradually set the context (e.g. deep learning, large datasets, label noise), before the specifics of the task are addressed. Overall, after adding a new paragraph on temporal localization and including Figure (1) based on your suggestions, we feel that this new introduction sets the stage for the rest of the paper more completely.
> Finally, we have made some clarifications in Section 2.
>
>
> (4) Evaluation Objective
> In all honesty, we are not sure to fully understand the suggestion. In the event that the modifications made in the introduction and in Section 3 and 4.1 do not answer this, would it be possible to clarify this point, so that we can improve the text accordingly?
>
>
> We hope that we have properly included your suggestions in the updated version of the text. If you have any other suggestions, please let us know, so we can take them into account.

---

### Official Review · AnonReviewer2 · 2019-10-24
**Official Blind Review #2**

**Rating:** 6

**Review:**

This paper deals with temporal label noise, or label misalignment problem in localization, which is the long-standing problem of robust event localization regarding the temporally misaligned labels in sequential data.  Some existing works are constrained to hardly fit the annotations of data during training. The work models the temporal label misalignment with perturbations to the ground-truth timestamps, and presents a soft localization learning scheme which relaxes the reliance on the exact position of labels. The idea introduced to temporal data is well-motivated since the label for temporal data is often dispersed. Technically speaking, the proposed SoftLoc loss comprises two terms: 1). $\mathscr{L}_{SLL}$: a soft learning loss that relaxes the prediction mass concentration, by symmetrically filtering the labels and predictions. This term helps to relax the model’s reliance on exact label locations. 2). $\mathscr{L}_{MC}$: a mass convergence loss that acts as a regularizer to facilitate the model with precise impulse-like localizations. With a trade of factor to balance two terms, the SoftLoc model can achieve precise impulse-like localization performance without weakening the model robustness. Various applications, such as PIANO ONSET, DRUM DETECTION and TIME SERIES DETECTION are performed to verify the effectiveness of the proposed method. And state-of-the-art performance is achieved.

Despite the achievement indicated by the experiments, I still have some concerns about this paper.

(1)	The authors propose the relaxed loss L_{SLL} for the soft learning of the location. And the label smoothing idea (applying a ˜S^2-Gaussian filter to the labels) has been introduced to increase the robustness to temporal misalignment of annotations [2]. Although the authors discuss the several inherent drawbacks of it in the 2nd paragraph of Related Work, they still follow the label smoothing idea in their model (Eq.2) in Section 4.1 in a two-side smoothed process. The reason should be clarified.

(2)	The authors claim that their method has the advantage of generalizing some regimes of weakly-supervised learning. And the adopted L_{MC} is a weakly-supervised loss. Does the advantage come from L_{MC}?

(3)	The authors propose the two-side relaxed loss L_{SLL} for the soft learning of temporal localization problem. However, in the ablation study, the authors do not give the results with different level of label noise for one-side variant. Adding these results could help to demonstrate the necessity of the two-side relaxed loss L_{SLL}.


References:
[1] Improved musical onset detection with convolutional neural networks
[2] Onsets and frames: Dual-objective piano transcription





**Experience Assessment:**

I have published one or two papers in this area.

**Review Assessment: Checking Correctness Of Derivations And Theory:**

N/A

**Review Assessment: Checking Correctness Of Experiments:**

I assessed the sensibility of the experiments.

**Review Assessment: Thoroughness In Paper Reading:**

I read the paper at least twice and used my best judgement in assessing the paper.

---

> ### Author Response · Authors · 2019-11-10
> **Response to Reviewer 2**
>
>
>
> Thank you for your constructive remarks. Based on your comments, we have updated Sections 3 and 4.1 to focus more on the choice of methodology and have added Figure 3 (and also Figures 2 and 4) to highlight even more clearly the difference between our model and the classical approaches.
>
>
> (1) Method Clarification
> We agree that the motivation could have been expressed more clearly. The text (Sections 3 and 4.1) has been updated to provide more clarity on the fundamental conceptual difference between our approach and the classical ones.
>
> In short, there is no issue with smoothing per say, problems arise when smoothing the labels **only**. However, most of these issues can simply be solved by not only smoothing the labels, but also the predictions. While this change might seem subtle, it has quite large implications in terms of modelling. A new figure (3) illustrates the main conceptual differences. Overall, instead of transforming the point prediction problem into a distribution prediction problem, smoothing both the labels and predictions allows the model to infer point predictions directly. It is only the loss function that views these point predictions as distributions and thus that deals with the temporal uncertainty of the labels. (See also discussion on novelty in the reply to Reviewer 1.)
>
>
> (2) Generalization of Past Models
> Obviously, as the proposed loss is a linear combination of a soft fully supervised loss ($L_{SLL}$) and a weakly supervised loss ($L_{MC}$), setting $\alpha=1$ would result in a weakly-supervised loss (i.e. $L_{MC}$).  However, the comment about the generalization of regimes from fully supervised to weakly supervised (in Section 4.3.) focuses on the power of the smoothing parameter $\mathcal{S}_{M}$. On the one side of the spectrum, setting $\mathcal{S}_{M}=0$ yields a loss that strictly takes the location of the labels into account (e.g. stepwise cross-entropy). One the other side, $\mathcal{S}_{M}\to\infty$ yields a loss that totally discard all prior location information (i.e. $L_{MC}$).  Therefore, varying $\mathcal{S}_{M}$ allows to control the amount of relaxation of the model reliance on the exact localization of the labels (ranging from fully supervised to fully supervised.) Overall, while the $\alpha$ parameter regulates the predominance of the mass convergence over the soft learning, the softness parameter can define a wide range of models ranging from a strict fully-supervised model to a weakly-supervised model.
>
>
> (3) Ablation Study
> Both benchmarks from Hawthorn et al. (Figure 1 in Section 5.1.1) are using one-sided smoothing with some peak picking. Their performance in moderate to high noise settings demonstrate the necessity for another approach. Additionally, the ablation study was designed to evaluate the subcomponents of our full model. Since peak picking is not part of our final approach, we chose not to include the Hawthorn et al. benchmarks in the ablation study. Second, one could further be interested in a model trained solely with one sided smoothing loss without peak picking. This approach is however bound to yield very poor results since the model would infer a distribution and the evaluation metric requires point predictions. Finally, the performance of a one-sided smoothing combined with the weakly supervised $L_{MC}$ could be assessed. However, these two losses have conflicting objectives, as the former requires the prediction to be spread out over time, while the latter requires the prediction to be point-like. The training of models with conflicting loss functions is very unstable and, in this case, would certainly lead to poor results. Overall, we preferred to only include the two components of the final loss function in our ablation study and feel that including weak benchmark (e.g. one sided without peak picking) would only be viewed as an attempt to artificially embellish our results.
>
>
> We hope that this completely answers your questions. If not, feel free to ask for more detailed explanations.

---

### Official Review · AnonReviewer1 · 2019-10-29
**Official Blind Review #1**

**Rating:** 3

**Review:**

The authors propose SoftLoc, an interpolation of two temporal event prediction losses (one based on label & prediction smoothing, and one based on weakly-supervised count-based loss) that makes the predictor more robust to noisy training data. They demonstrate on various temporal alignment datasets from music to wearable sensors to video action segmentations, that their method is performs well both on noisy-free and noisy settings compared to prior approaches.

Strengths:
(1) relatively thorough experimental valuations: using 4 datasets comparing with sufficient number of prior approaches (One potential improvement could be to try noise distributions other than Gaussian)
(2) simple objective and consistent improvements. It is encouraging that the simple modification enables such consistent empirical improvements.

Weaknesses:
(1) the novelty of the method appears limited. The weakly-supervised loss is borrowed from the prior work, so it seems like the main algorithmic novelty is to add noise to predictions (as opposed to just adding to labels as done in prior work).

Other comments:
(1) in 5.1.1, is there justification for \alpha_\tau expression? How did you pick that?
(2) is Issue 1 an actual
 problem? Even with just label smoothing, I would expect predictor will find the mode and peak-picking can produce the correct predictions. Similarly, discussions of how the proposed method can solve the issues listed may benefit from some rewriting, or even simple toy experiments to demo that those are actually the concerns that are not addressed by prior work. Such toy experiments may complement the existing end-to-end experiments to demonstrate the precise properties of the proposed method.

**Experience Assessment:**

I do not know much about this area.

**Review Assessment: Checking Correctness Of Derivations And Theory:**

I assessed the sensibility of the derivations and theory.

**Review Assessment: Checking Correctness Of Experiments:**

I assessed the sensibility of the experiments.

**Review Assessment: Thoroughness In Paper Reading:**

I made a quick assessment of this paper.

---

> ### Author Response · Authors · 2019-11-10
> **Response to Reviewer 1 (part 2/2)**
>
>
>
> ---- Other comments
>
> (1) The general motivation for the choice of $\alpha_{\tau}$ is briefly mentioned in Section 4.2.: “From a practical standpoint, starting with a moderate $\alpha_{\tau}$ allows an initial relaxed localization learning, before performing stronger mass convergence”. More precisely, this setting allows for an initial long (>1e5 iterations) learning phase with $\alpha_{\tau}=0.2$, where the approximate location can be learnt without much consideration for the sparsity of the predictions. The core of the representation learning is also achieved during this phase. After this stage, the predictions are generally scattered over several time-steps. The $\alpha_{\tau}$ is then linearly increased until obtaining precise point predictions. Almost no hyperparameter tuning was performed in this paper as the central objective of the work is about robustness to noise and not raw performance. We simply ensured that 1) $\alpha_{\tau}$ is initially small enough for the smooth location learning loss to drive the training, 2) the initial phase lasts long enough for the location learning loss to plateau and 3) the number of iteration is large enough to allow for the convergence of the full loss.
>
>
> (2) Issue 1 is certainly a problem in moderate to high noise settings as demonstrated by the experiments. In Section 5.1.1. the benchmarks from Hawthorne et al. performs label smoothing and peak picking. While their performance is overall very good when the standard deviation of the normal noise is under 25ms, their score plummets after that point.  In noisy settings, the main issue comes from the fact that the performance of such approaches relies heavily on peak picking, which is not part of the learning process itself. In contrast, our approach learns precise localization in an end-to-end fashion.
>
> Following your recommendation, Sections 3, 4.1, 4.2 have been rewritten to explain in more details the benefits of our model against classical approaches, while illustrations (Figures 2 and 4) have been added to the text for a more visual presentation of the model’s properties.
>
>
> We hope we were able to address your concern. If you have any further concern or question, do not hesitate to ask.

---

> ### Author Response · Authors · 2019-11-10
> **Response to Reviewer 1 (part 1/2)**
>
>
>
> Thank you for your review and helpful feedback. We hope that the following clarifications on novelty, noise distribution and parameter settings will address your questions and concerns, while the newly added discussion (Sections 3, 4.1 and 4.2) and figures (2, 3 and 4) provide some more insight into the properties of our model.
>
> ---- Novelty
>
> The paper presents a novel paradigm for dealing with temporal uncertainty. We agree that the initial version of the paper only briefly mentioned this important point. Therefore, the updated version of the text contains a more focused discussion on the novelty of our approach (see Sections 3, 4.1 and 4.2) as well as a diagram (Figure 4) illustrating the main conceptual difference with previous models.
>
> In short, the classical approach for dealing with label misalignment consists in first transforming the point labels into distributions through smoothing or other transformations [1,2]. The algorithm is then trained to predict these distributions, that eventually must be transformed back to point prediction using hand-crafted peak picking heuristics. This methodology of transforming point prediction into distributions and inferring full resolution heatmaps rather than points is also very common in image keypoint detection application that must deal with spatial uncertainty, e.g. human pose estimation [3,4] or facial landmark detection [5].
>
> In contrast, the proposed loss function allows to train models to directly infer point predictions without having to explicitly deal with distributions or heatmaps. The loss function itself models the temporal uncertainty. Such a direct approach presents several main advantages over the classical approach. For instance, our loss allows for end-to-end learning of localization without having to resort to additional heuristics. Additionally, as mentioned in the paper, label smoothing presents several issues (left-tail estimation, disentanglement of close predictions, …) which are solved by our approach (see new Figures 2 and 4). These differences are further exacerbated in complex noisy settings.
>
> This general approach could find applications beyond the temporal domain. For instance, a 2D continuous version of the proposed loss function could be leveraged to successfully train offset detection models in ambiguous or noisy location settings, without having to resort to full resolution heatmap predictions [3,4,5].
>
> We consider the simplicity of our approach a strength rather than a flaw, especially given its competitive results on various experiments.
>
> [1] Improved musical onset detection with convolutional neural networks (ICASSP 2014)
> [2] Onsets and frames: Dual-objective piano transcription (ISMIR 2018)
> [3] Efficient Object Localization Using Convolutional Networks (CVPR 2015)
> [4] Joint training of a convolutional network and a graphical model for human pose estimation
> [5] Robust Facial Landmark Detection via a Fully-Convolutional Local-Global Context Network (CVPR 2018)
>
>
> ---- Noise distributions
>
> Other noise distributions are already included in the paper. Indeed, the time series detection experiment (Section 5.2) is carried out not only with Gaussian noise but also with binary constant length shifting of labels. Similarly, the label sequences in the video experiment (Section 5.3) are either delayed or advanced by a fixed constant.
> However, we concede that all the tested noise distributions were centered and non-skewed. Therefore, to expand the range of noise distribution for the reader, we have conducted once again the wearable sensors experiment (Section 5.2) with a skew normal distribution of the noise ($alpha=-2$). The results confirm once again the effectiveness of the model to deal with label misalignment, even in skewed non-symmetrical settings:
>
> -------------------------------------------------------------------------------------
> LR-M       |  79.7 (10.4)  |  68.3 (15.6)  | 61.4 (20.7)  |  54.7 (18.2)
> -------------------------------------------------------------------------------------
> CE            |  57.6 (16.6)  |  27.8 (13.7) |  20.0 (13.9)  |  16.1 (14.4)
> -------------------------------------------------------------------------------------
> SoftLoc   |  90.4 (3.9)    |  88.2 (5.0)    |  84.2 (6.1)    |  79.1  (9.0)
> -------------------------------------------------------------------------------------

---

### Author Response · Authors · 2019-11-14
**Improvements**



We want to thank again the reviewers for taking the time to review our paper. We have taken the reviewers’ valuable remarks to heart and have made the following modifications to the text: (As our response to the reviewers has not sparked any additional comment nor discussion so far, all modifications are based on the initial reviews.)

(1) *Motivation and clarification of the choice of methodology* (R1, R2, R3). Sections 3 and 4.1 more thoroughly explain the main conceptual differences between our proposed method and classical approaches. In addition, Figures 2-4 have been created and added to illustrate these differences (Figure 3), the main drawbacks of the classical approach (Figure 2) and how they are solved by our methodology (Figure 4). A comment about the end-to-end nature of our approach has also been included in Section 4.2. (See also the respective responses to the reviewers for full discussion).

(2) *The Introduction and Related Works Sections* focus more extensively on the specific task addressed by this paper (R3). A full paragraph introducing the task of temporal event localization under label misalignment and Figure 1 have been added to introduce more directly to the reader the problem solved in this work. In addition, the Related Works Section has been restructured to be more centered around the problem of temporal label misalignment.

(3) *The novelty* of the proposed approach has been highlighted more clearly in Section 3 and Section 4.1 (R1). The newly created Figure 3 helps to visualize the novel modelling paradigm.

(4) The experiments in Section 5.2. have been reconducted using a *skew normal distribution of noise* (R1). A comment has been added to inform the reader that the results are consistent with those of non-skewed distributions.

Other minor modifications have been made.


We hope that these modifications fully reflect all the reviewers’ comments and concerns. If there are any further suggestions left that could improve the submission, we will be more than happy to incorporate them in the final version of the text.

---

### Author Response · Authors · 2019-11-15
**Authors’ Final Remarks**



We would like to take this final opportunity to thank again the reviewers for their comments and suggestions. Their reviews have allowed us to update and improve the text, especially in terms of clarity and motivation (see “Improvements” and reviewers’ specific answers).

For practical applications, where perfect annotations of sequential data cannot be guaranteed, having access to a robust and simple alternative to the extensively used cross-entropy is crucial. This work thus proposes an intuitive yet effective solution to this understudied problem. The introduced loss displays strong empirical results by achieving consistent and significant improvement over state-of-the-art benchmarks in 4 tasks from 3 different domains (audio, wearable sensor time series, video), (codes fully available).

We believe that, after the thorough textual improvements suggested by the reviewers, this submission which combines a sensible solution to an important problem and an extensive empirical evaluation is of scientific interest to the ICLR community.

---

### Decision · Program_Chairs · 2019-12-19

**Decision:**

Reject

**Comment:**

Main content:

Blind review #3 summarizes it well:

This paper proposes a new loss for training models that predict where events occur in a sequence when the training sequence has noisy labels. The central idea is to smooth the label sequence and prediction sequence and compare these rather than to force the model to treat all errors as equally serious.

The proposed problem seems sensible, and the method is a reasonable approach. The evaluations are carried out on a variety of different tasks (piano onset detection, drum detection, smoking detection, video action segmentation).

--

Discussion:

The reviewers were concerned about the relatively low level of novelty, simplicity of the proposed approach (which the authors argue could be seen as a feature rather than a flaw, given its good performance), and inadequate motivation.

--

Recommendation and justification:

After the authors' revision in response to the reviews, this paper could be a weak accept if not for the large number of stronger submissions.